# New Trends and Advances in Non-Variceal Gastrointestinal Bleeding—Series II

**DOI:** 10.3390/jcm10143045

**Published:** 2021-07-08

**Authors:** Pablo Cañamares-Orbís, Ángel Lanas Arbeloa

**Affiliations:** 1Gastroenterology, Hepatology and Nutrition Unit, San Jorge University Hospital, 22004 Huesca, Spain; 2IIS Aragón, CIBERehd, 50009 Zaragoza, Spain; alanas@unizar.es; 3Service of Digestive Diseases, University Clinic Hospital Lozano Blesa, 50009 Zaragoza, Spain; 4University of Zaragoza, 500009 Zaragoza, Spain

**Keywords:** gastrointestinal bleeding, peptic ulcer bleeding, Helicobacter pylori, NSAIDs, PPI, Glasgow-Blatchford, colonic bleeding, diverticular bleeding, small bowel bleeding, capsule endoscopy, endoscopic therapy, antithrombotic management

## Abstract

The gastrointestinal tract is a long tubular structure wherein any point in the mucosa along its entire length could be the source of a hemorrhage. Upper (esophagel and gastroduodenal) and lower (jejunum, ileum, and colon) gastrointestinal bleeding are common. Gastroduodenal and colonic bleeding are more frequent than bleeding from the small bowel, but nowadays the entire gastrointestinal tract can be explored endoscopically and bleeding lesions can be locally treated successfully to stop or prevent further bleeding. The extensive use of antiplatelet and anticoagulants drugs in cardiovascular patients is, at least in part, the cause of the increasing number of patients suffering from gastrointestinal bleeding. Patients with these conditions are usually older and more fragile because of their comorbidities. The correct management of antithrombotic drugs in cases of gastrointestinal bleeding is essential for a successful outcome for patients. The influence of the microbiome in the pathogenesis of small bowel bleeding is an example of the new data that are emerging as potential therapeutic target for bleeding prevention. This text summarizes the latest research and advances in all forms of acute gastrointestinal bleeding (i.e., upper, small bowel and lower). Diagnosis is approached, and medical, endoscopic or antithrombotic management are discussed in the text in an accessible and comprehensible way.

## 1. Introduction

Gastrointestinal bleeding (GIB) is a common emergency, and a frequent cause of hospitalization in gastrointestinal, internal medicine, or surgery departments. Esophagogastroduodenoscopy and ileocolonoscopy are fundamental tools to research the source of hemorrhages. More recently, physicians have been able to explore the small bowel by capsule endoscopy and enteroscopy. Therefore, we can explore endoscopically the entire GI tract. However, there are still GIB events whose source cannot be determined and are described as “obscure GIB”.

The incidence of upper GIB has decreased progressively in the last few decades, whereas lower and obscure GIB have either increased slightly or remained stable [1,2]. The widely use of proton-pump-inhibitors (PPI) and the extended investigation and eradication of Helicobacter pylori infection have been pointed out as the main reason for the observed decline in the incidence of non-variceal upper GIB [3]. However, PPI do not protect the lower gastrointestinal tract and may induce changes in the microbiota of the small bowel and colon which could render the mucosa more susceptible to damage induced by NSAIDs or aspirin [4]. Diverticular bleeding is the most common cause of lower GIB, and diverticular disease is more prevalent in the elderly who usually have more comorbidities and often take anticoagulant or antiplatelet drugs [3].

Mortality rates associated with GIB remain high and are typically between 5–10% in upper GIB [5,6] and 3.9% in lower GIB [7]. Mortality risk could also increase after hospital discharge. A study showed three times higher mortality in patients after an upper GIB event during the 32 months follow-up period [8]. Factors predicting mortality include old age (>80 years), renal failure, liver cirrhosis, advanced malignancy, low hemoglobin (<7 g/dL) on admission, and failed endoscopic hemostasis [9].

This article records the latest innovations in upper, lower, and obscure (small bowel) GI bleeding.

## 2. Upper Gastrointestinal Bleeding

Peptic ulcers remain the first cause of upper GIB (about 47% of all cases) [10]. Other causes include gastritis (18%), esophagitis (15%), angiodysplasia (6%), Mallory-Weiss (7%), neoplasm (3.7%), esophageal varices (1.8%), and Dieulafoy’s lesion (1.5%). A decrease in incidence is supported by a diminution of gastritis and peptic ulcer hospitalization rates [10]. On the other hand, neoplasm, esophagitis, and angiodysplasia origin have slightly increased, from 20% to 50% between 2002 and 2012 [10].

Hematemesis and melena are the main signs of upper GIB. Hematemesis refers to vomiting of either red blood or coffee-ground emesis and suggests bleeding proximal to the ligament of Treitz. Melena is defined as black, tarry stool that occurs several hours after the bleeding event and results from the degradation of blood to hematin or other hemochromes by gut bacteria [11]. Hematochezia refers to red or maroon blood in the stool. It is usually observed in lower GIB, but is sometimes caused by upper GIB (mainly associated with hemodynamic instability).

### 2.1. Risk Factors

Nonsteroidal anti-inflammatory drug (NSAIDs) use, including low-dose aspirin taken for secondary or primary prophylaxis of cardiovascular diseases, and Helicobacter pylori infection are the main risk factors for development of peptic ulcer disease [12]. H. pylori is a bacterium present in approximately half of worldwide population which has adapted to an acid environment [13]. Chronic inflammation through gastritis is the mechanism to facilitate gastric and duodenal ulcer formation. Pangastritis decreases pH secretion and is associated with gastric ulcers, whereas antrum gastritis increases pH secretion and is associated with duodenal ulcers [5]. In addition to topical action, NSAIDs contribute to mucosal damage by inhibiting cyclooxygenase-1 (COX-1). COX-1 allows for the formation of prostaglandins which play a protective role in the gastrointestinal mucosal barrier by stimulating mucous and bicarbonate secretion, inhibiting acid secretion, and promoting cell proliferation and mucosal blood flow. NSAIDs cause a low-grade “ischemic effect”, especially in gastric mucosa, damaging blood vessels and forming ulcers [5]. The harmful effect of NSAIDs together with H. pylori infection increases the chances to develop a peptic ulcer and related bleeding complications.

Non-aspirin antiplatelet and anticoagulant drugs do not harm the gastrointestinal mucosa directly. However, they facilitate bleeding from pre-existing or new lesions [14].

However, approximately 20% of peptic ulcer are non-drugs or H. pylori-related [15]. Older age, mesenteric ischemia, smoking, and the presence of other concomitant diseases have been raised as other risk factors for these “idiopathic” ulcers [16]. These ulcers have higher rebleeding and mortality rates according to a study performed in Hong Kong [17].

### 2.2. Prognosis Scores

All clinical guideline recommends the use of scores to lead the management of upper GIB patients in Emergency Departments (Appendix A).

The main scores for upper GIB are Glasgow Blatchford (GBS), AIMS65, PNED, admission Rockall, and full Rockall. GBS, AIMS65 and admission Rockall are pre-endoscopic scores, whereas PNED and full Rockall include endoscopic outcomes. Each score was designed for a defined outcome, so they show different aspects among them. For example, the GBS score was designed to predict need of intervention (blood transfusion, endoscopic therapy, surgical intervention, etc.), whereas AIMS65 was designed to predict mortality and length of hospital stay. More recently, the ABC score has been developed to predict mortality in upper and lower GIB. It includes age, urea, albumin, creatinine levels, and comorbidities such as altered mental status, liver cirrhosis, disseminated malignancy and ASA score. Patients can be classified into three groups: low risk if the patient scores ≤3 points; moderate risk from 4 to 7 points; and high risk if ≥8 points. ABC score has been compared with previously existing scores to predict mortality in upper and lower GB, showing greater specificity and area under curve (AUC) in both cases [18].

Classic five scores (admission Rockall, AIMS65, and Glasgow Blatchford) and post-endoscopy scores (full Rockall and PNED) were compared in an important and multicenter prospective study in over 3000 patients. The study concluded that a GBS score ≤1 represents the optimum low-risk threshold for outpatient management [19]. GBS only misclassifies <1% of high-risk patients as low risk [20], so this score can be useful in deciding the safe discharge in an emergency department. A score ≥7 has the highest sensitivity and specificity for need of endoscopic treatment [21]. Likewise, intermediate-risk ABC score patients have a 7% risk of mortality, which increases to 25% in high-risk patients [18].

### 2.3. Medical Management before Endoscopy

Resuscitation is the first and most important intervention when a patient suffering an upper GIB arrives to the hospital. Some studies have compared which fluid therapy is better, not specifically in GIB but in critically ill patients. No significant differences have been shown between the use of crystalloid or colloid solutions [22]. Airway protection must be mandatory in patients with decreased level of consciousness to prevent bronchoaspiration, so tracheal intubation may be an option.

A complete patient history asking about antiplatelet, anticoagulant, and non-steroidal anti-inflammatory medications is essential for the interview. Furthermore, questions to detect the origin of the bleeding are also essential (knowing that massive upper GIB could manifest as rectorrhagia) usually when hemodynamic instability is present. The next step should be to perform blood tests to assess the main biochemical parameters, hemoglobin, and coagulation level.

A restrictive transfusion (hemoglobin < 7 g/dL) policy for patients without comorbidities has been well established due to the reduction in all-cause mortality compared to liberal transfusion [23,24]. However, a higher cut-off level is recommended for transfusion in patients with cardiac comorbidities (e.g., previous myocardial infarction, heart failure, instability, etc.), with transfusion being indicated when the hemoglobin level falls below 8 g/dL. Despite these general instructions, it is important to know that transfusion should be guided for patient’s clinical status and not only for hemoglobin level (e.g., when massive bleeding is suspected) [20].

PPI therapy must be prescribed when a peptic ulcer is suspected. Its objective is to facilitate the clot in the case of peptic lesions. In vitro studies have shown that coagulation and platelet aggregation are not functional when pH is inferior to 5.9 [5]. Studies on patients have shown that PPI therapy achieves a reduction in high-risk stigmata ulcers when endoscopy is performed, although no differences were found related to rebleeding, mortality, or need of surgery [25,26]. However, the optimal dose of PPI remains controversial. Clinical consensus still recommends 80 mg intravenous bolus followed by 72 h of 8 mg/h continuous intravenous infusion, which is considered to be a high-dose therapy [20]. Some studies suggest that low-dose therapy may be equally effective in preventing rebleeding, need of surgery, and mortality [27,28]. Thus, no strong recommendation can be made for this PPI dose.

Tranexamic acid is an antifibrinolytic drug used for pulmonary bleeding and it has been suggested to be useful in GIB (both upper and lower). However, new data from a large randomized controlled trial (HALT-IT) with about 12,000 patients are recently available showing the ineffectiveness of tranexamic acid in patients with upper and lower GIB [29].

### 2.4. Diagnosis

Esophagogastroduodenoscopy is the gold standard method for confirming an upper GIB and diagnosing its cause. When doubts exist, physician used to place a nasogastric tube in stomach. Gralnek et al. [30] found that capsule endoscopy was safer and more convenient than nasogastric aspiration in identifying the source of bleeding in a prospective cohort study. It could be useful to confirm in doubtful cases of GIB, but guidelines do not recommend capsule endoscopy use in an extensive way since more studies are needed to establish this role [31] and to justify its expensive cost.

Another challenging question is the best time to perform an endoscopy. A recent randomized controlled trial from Prince of Wales Hospital (Hong Kong, China) has shown no difference in rebleeding and 30-days mortality between performing endoscopy within 6 h or within 24 h after the endoscopist consultation [32]. The study includes patients with GBS greater than 12 (high-risk patients) and patients with variceal bleeding. All Asian, European, and American guidelines supported this recommendation previously [33,34,35], but this study provides more evidence.

The need for routine second-look endoscopy within the first 24 h after the index urgent endoscopy was widely debated some years ago. This method is not a cost-effective procedure according to recent data [36]. More recently, no differences in surgery, radiological intervention, or mortality have been shown in a randomized prospective study between the two options. Although no statistically different, the rebleeding rate was superior in the second-look group (10.2% vs. 4.5%, *p* = 0.13) [37]. However, second-look endoscopy may be necessary if rebleeding is suspected because of an increasement in transfusional requirements, clinical data, or with some specific etiologies such as ischemic ulcers.

### 2.5. Endoscopic Treatment

Endoscopic treatment is mandatory when high risk stigmata of bleeding are found. Forrest classification continues to be the gold standard for classifying peptic ulcer bleeding. Active peptic ulcer bleeding (Forrest Ia and Ib) and visible vessel ulcers (Forrest IIa) must be treated endoscopically [38]. Adherent clot ulcers (Forrest IIb) may be also treated after clot removal [33]. Flat pigmented spot (Forrest IIc) and a clean base ulcer (Forrest III) do not need endoscopic treatment, and patients can be safely discharged with PPI therapy if they do not suffer another comorbidity [38].

Endoscopy ultrasound (EUS) and the Doppler effect might help diagnosis in upper GIB. The aim is to detect the arterial flow in the ulcer base by a Doppler probe that it is introduced through the working channel. Some authors point out that high risk stigmata ulcer is better assessed with Doppler ultrasound endoscopy in comparison to simple gastroscopy, especially in Forrest IIa and Forrest IIc ulcers, where more variability in Forrest classification exists between endoscopists [39]. Moreover, Doppler study permits one to check the flow after an endoscopic treatment. Up to 89% of incomplete treated lesions will rebleed in the follow-up period [40]. A randomized trial studied differences between patients with normal management and patient with Doppler-probe guide treatment. The rebleeding rate at 30 days was significantly lower in the Doppler probe group (26.3% vs. 11.1%; *p* = 0.0214) [39]. In addition, a Doppler-probe study has shown that Forrest Ib ulcers, previously considered as high-risk ulcer for rebleeding, have actually very low rate of rebleeding [41]. EUS has been used to introduce coils or cyanoacrylate (“a special glue”) in gastric or duodenal varix [42]. A major limitation of EUS is the limited number of endoscopists trained in this method. Currently, guidelines do not recommend its use in upper GIB.

Endoscopic treatment for upper GIB is based on four pillars: injection, thermal coagulation, mechanical therapy, and topical therapy. The most effective endoscopic treatment to achieve hemostasis in peptic ulcer bleeding is adding a second method (thermal or mechanical) to diluted epinephrine (1:10,000) injection, according to two metanalyses of controlled trials [43,44]. Mechanical treatment refers to through-the-scope clips (endoclips), although some new advices like over-the-scope clips have been developed recently (this will be discussed later in the manuscript). Thermal therapy includes contact and non-contact methods. Several probes are available for contact therapy such as multipolar, heater probe, and monopolar probes. Non-contacts therapy can be applied by argon plasma. However, thermal therapy is not always available, and clips are sometimes difficult to shoot depending on the anatomic situation of the bleeding point. Sclerosant agent injection could be a good alternative therapy. The efficacy of argon plasma seems to be similar to sclerosant agents or heater probse according to a metanalyses of randomized trials [33,45,46]. Argon plasma therapy is more frequently used for angioectasias in both the upper and lower gastrointestinal tracts [47]. 

Table 1 summarizes pre and post-endoscopy management in peptic ulcer bleeding depending on endoscopic stigmata.

New tools are emerging to help endoscopists with the control of GIB. These tools are mechanical similar to over-the-scope clip (OTSC; Ovesco, Tübingen, Germany), endoscopic suturing or band ligation, topicals such as Hemospray^®^ (Cook Medical Inc., Bloomington, IN, USA) or cryotherapy, and thermal like radiofrequency ablation.

#### 2.5.1. Over-the-Scope Clips

A recent randomized clinical trial compared OTSC with standard therapy in recurrent bleeding for peptic ulcer, although with a limited number of patients [48]. While this study reported a decrease in persistent bleeding, no differences were found in need of surgery and mortality rates. Previously, only some cases series had been reported showing that OTSC was useful in large ulcers up to 5 cm [49], with a successful rate close to 80% achieving hemostasis after a rebleeding event [50]. OTSC is considered a good alternative for refractory ulcers with difficult control by classic management, but it has also been described as first line in bleeding peptic ulcers [51] and Dieulafoy’s lesions or bleedings after gastric polypectomy [52]. OTSC was recommended to treat refractory upper GIB in the European guidelines in 2015, and in the Asian-Pacific working group in 2018 [33,35].

#### 2.5.2. Hemospray^®^

Hemospray is composed of TC-325, a mineral-based hemostatic powder, applied from a working channel. It is safe and completely eliminated from the gastrointestinal tract after 70 h [53]. It is useful to achieve temporal hemostasis as bridge to a definitive therapy [20]. Hemospray therapy achieved a similar rate of primary hemostasis in comparison with mechanical endoclips [54]. However, Haddara et al. found an immediately efficacy rate of 96% in a large multicenter study, with high recurrence rates on day 8 (26.7%) and day 30 (33.5%) [53]. This therapy seems especially useful in cases of diffuse bleeding (e.g., neoplasm origin) compared to unique-point bleeding (e.g., peptic ulcer or Dieulafoy’s) [40,55]. The single use of Hemospray seems to cause more rebleeding than conventional therapy [54,56], but both therapies applied together may decrease the costs due to a reduction in the rebleeding rates [57]. Other systems similar to Hemospray have been presented and have promising outcomes but have not been studied as much as Hemospray. EndoClot (EPI, Santa Clara, CA, USA) and Ankaferd Blood Stopper (ABS) (Ankaferd Health Products, Istanbul, Turkey) [58] represent some of these new compounds.

#### 2.5.3. Endoscopic Suturing

Firstly, used for fistulas, leaks and perforations, these novel dispositive needs experienced endoscopists. However, the technique is promising. Few patients have been treated so far with this procedure, which seems to be another alternative for refractory upper GIB. Success rates of 100% and no rebleeding within 72 h in ten patients with gastric or duodenum ulcers have been reported [59]. It is based on the system OverStitch™ (Apollo Endosurgery, Austin, TX, United States) which consist of a cap-based suturing system with a curve suture arm and another anchor exchange arm [60]. The system must be introduced with an over-tube. The disadvantages of this method are necessary previous training of endoscopist and a required double-channel endoscope. Malignancy should be excluded previously [61] and it is more beneficial for marginal ulcers in anastomosis locations [62].

#### 2.5.4. Band Ligation

This mechanical tool is already used for esophageal varix bleeding. Band ligation have been successfully described in Dieulafoy’s lesion and gastric antral vascular ectasia (GAVE). No differences were observed between band ligation and endoclip in Dieulafoy’s lesion in a randomized prospective clinical trial [63]. In patients treated with either argon plasma or band ligation, Keohane et al. [64] reported endoscopic improvement of GAVE lesions with band ligation. However, no differences were observed in other parameters such as hemoglobin level and transfusions. Zepeda-Gómez et al. [65] reported a clinical response of 91% with a significant improvement of hemoglobin levels and number of transfusions requirements per month in a case series study. Both therapies (argon plasma and band ligation) for these indications and a lack of data are not enough to recommend one over the other.

#### 2.5.5. Cryotherapy

It induces cell necrosis through localized freezing in a tissue. Cryotherapy has been proposed as treatment for GAVE in patients in whom argon plasma coagulation has previously failed. A pilot study showed that it is a safe and effective therapy, with completely resolution in 50% of patients, and a partial response in the other half in three endoscopic sessions [66]. More studies are necessary to know the true clinical application with this therapy in patients with GAVE [61]. Furthermore, the system is not available in most endoscopic centers worldwide [67].

#### 2.5.6. Radiofrequency Ablation (RFA)

RFA has been widely applied in Barrett esophagus treatment with and without dysplasia [68]. RFA has been proposed as a good alternative for argon plasma in GAVE lesions. It could be applied with a large plate which encompasses more tissue surface, making the procedure more comfortable. In a systematic review with a relatively large number of patients (72 patients), 74% of patients with RFA achieved a clinical response with only 4.2% of non-fatal adverse effects reported [69].

#### 2.5.7. Coagrasper (Olympus Corp., Tokyo, Japan)

This endoscopic devide, which is introduced through the working channel, combines thermal and mechanical hemostasis. It works at a lower voltage, being associated with a lower risk of perforation due to less damage in deep tissue [70]. The forceps was developed to reduce bleeding after endoscopic submucosal dissection (ESD). This hemostatic forceps has been compared with both endoclips and heater probes in randomized clinical trials of peptic ulcer bleeding, ultimately detecting better control of hemorrhage with the Coagrasper [71,72].

### 2.6. Antiplatelet and Anticoagulant Management

The use of antiplatelet and anticoagulants drugs has been increasing as aging populations grow. They are typically prescribed for ischemic heart disease, cerebral thromboembolic disease, deep venous thrombosis, pulmonary embolisms, peripheral thrombosis, and other situations. In addition, low-dose aspirin (LDA) can be used for primary cardiovascular prophylaxis in high-risk patients. Physicians face a difficult clinical situation, since they need to balance the risk of CV events against the risk of prolonged or recurrent bleeding, depending on the clinical decision of interrupting or maintaining of the antithrombotic drug. Resumption of therapy after drug interruption is another clinical decision that needs to be considered. Early resumption of the antithrombotic therapy has been associated with a reduction in mortality and vascular events despite an increasement of rebleeding rates in cases of upper GIB [73]. In the case of lower GIB, the evidence is much lower and only antiplatelet therapy has been associated with an increased risk of rebleeding without differences in mortality [74].

Below is a summary of European, American, and Asian guidelines of the management of antithrombotic drugs in acute non-variceal upper GI bleeding [33,75,76].

#### 2.6.1. Use of a Single Antiplatelet Agent: (Mainly LDA, Sometimes Clopidogrel)

You can see in Table 2 where is single antiplatelet agent management in non-variceal upper GIB.

○Primary prophylaxis is to stop the drug. Discuss with the patient the benefits and risks of reintroducing the antiplatelet after the control of the bleeding.○Secondary prophylaxis is to stop the drug and resume within five days after the endoscopic hemostasis is achieved [76]. A second-look endoscopy might be considered to ensure the situation [33]. LDA might be continued in cases of mild upper GIB after discussing it with the patient.

#### 2.6.2. Dual Antiplatelet Agents: (LDA Plus Clopidogrel, Ticagrelor or Prasugel) 

This situation is frequent within the first year after an acute cardiovascular event (Table 3). The timing and type of coronary stents implanted may influence the decision. Cardiologist consultation is important. These drugs irreversibly inhibit platelet function, but no platelet transfusion is recommended in patients with upper GIB [76,77].

#### 2.6.3. Vitamin-K Antagonist: (Warfarin or Acenocumarol) 

They have been used widely, although currently, they are gradually deprecated in favor of direct anticoagulants, which do not need INR control, have a rapid onset of action, and have less drug interactions. Vitamin-K antagonist has modified coagulation through the INR (Table 4). A systematic review revealed that INR at presentation does not predict recurrent upper GIB [78], but many retrospective studies have shown a high success rate of endoscopic hemostasis with an INR between 1.5 and 2.5 [76]. Urgent endoscopy should not be delayed normalizing the INR, but anticoagulation converter drugs are recommendable in cases of supratherapeutic INR. Reversal effects of warfarin can be obtained with the administration of 5–10 mg intravenously of vitamin K for mild hemorrhages. However, prothrombin complex concentrates (PCC) are preferred for urgent reversal.

#### 2.6.4. Direct Oral Anticoagulants (Apixaban, Rivaroxaban, Dabigatran, and Edoxaban) 

These drugs inhibit certain clotting factors, namely thrombin (dabigatran) and factor Xa (apixaban, rivaroxaban and edoxaban) (Table 5). Drug dosage must be modified in patients with renal or hepatic impairment. The INR values are not modified by direct oral anticoagulants, and other methods to detect their effect are not widely available in emergency departments [79]. The half-life of the drug is about 12 h in patients without renal insufficiency, and the anticoagulation action is achieved quickly between the first 1 and 4 h [76]. Idarucizumab is the only available antidote, and is only effective for dabigatran. Another option when drug ingestion has occurred in less than 3 h is activated charcoal. Antagonists to anticoagulants that inhibit factor Xa, such as andexanet alpha, will be available soon.

## 3. Lower Gastrointestinal Bleeding

Hospitalizations due to lower GIB is becoming more frequent than upper GI bleeding [1]. Patients who suffer lower GIB tend to be older and have more comorbidities than patients with upper GIB, with similar rates of anticoagulant and antiplatelet drug use [80]. Treatment with NSAIDs, antiplatelet, and anticoagulant agents increase the risk of both upper and lower GIB [81].

### 3.1. Lower GIB Causes

Diverticular bleeding is the most frequent cause of lower GIB (26–33%) [80]. Other causes are ischemic colitis (16%), inflammatory bowel disease (11.7%), hemorrhoids (10.4%), colorectal cancer (7.4%), and arteriovenous malformations (3.1%) [82]. Other less common causes include post-polypectomy bleeding, solitary rectal ulcers, colitis induced by radiotherapy, etc.

### 3.2. Medical Management before Endoscopy

The initial management of moderate-serious lower GIB does not differ much of that described in upper GIB (Appendix A). Hemodynamic stability should be evaluated first, and resuscitation with crystalloids or colloids should be used to maintain an adequate blood pressure (if needed). No specific treatment is available to stop the hemorrhage. Fortunately, the majority of lower GIBs are self-limited, and we must ensure hemodynamic stability, blood replacement, and investigate their etiology.

Recommendations concerning blood transfusions in lower GIB are based on studies from upper GIB [83,84]. A restrictive blood transfusion is recommended when hemoglobin decreases from 7 g/dL, with the exception of patients with heart and cerebrovascular diseases who must receive transfusions to maintain hemoglobin above 8 g/dL.

### 3.3. Diagnosis

Colonoscopy is the preferred procedure, but its implementation is more complex than gastroscopy in upper GIB due to the need of bowel preparation after an adequate resuscitation. Diagnostic yield for colonoscopy ranges from 42 to 90%. This variation is due to the lack of standardization in the reporting of hemorrhagic findings in different studies [83]. The optimum time to perform a colonoscopy has remained uncertain. Strate et al. [85] showed that endoscopic therapy could be applied in 29% of colonoscopies performed within the first 12 h, whereas it should be 0% in colonoscopies performed after 48 h. However, a multicenter randomized trial in Japan has been recently published comparing colonoscopies within 24 h and colonoscopies between 24 and 96 h after hospital admission. No differences in hemorrhage stigmata identification, rebleeding within 30 days, length of stay, transfusions, or death were observed [86]. Other non-randomized study supported these results [83]. Currently, the American guideline from 2016 still recommends colonoscopies within 24 h of patient presentation [84], although more recent British guidelines do not make any recommendation about the optimal time for colonoscopies [83]. Unlike the recommendation of performing a gastroscopy within 24 h of patient admission with an upper GI bleed, it is unclear that the same timing for colonoscopies can be recommended in cases of acute lower GIB.

For patients with rectal bleeding and persistent hemodynamic instability, an upper GIB should be ruled out and an upper GI endoscopy can be performed first. Patients with hemodynamic instability may not support a colonoscopy procedure and should be avoided. Finding the bleeding source is of paramount importance and a computed tomography angiography (CTA) is the best option in this situation to plan the correct treatment [83]. CTA has a good sensibility and specificity (79–95%/95–100%, respectively) in these circumstances and could detect velocity bleeding up to 0.3–1 mL/min [83].

### 3.4. Prognosis Scores

In comparison to upper GIB, fewer validated scores for lower GIB have been developed to lead the management of these events in emergency departments. Firstly, Strate et al. analyzed some early predictors for severe lower GIB including tachycardia, low systolic blood pressure, presentation with syncope, non-tender abdominal examination, rectal bleeding within the first 4 h, aspirin use, and a Charlson score more than 2 [87]. The risk of severe lower GIB, understood as continued bleeding within first 24 h and recurrent bleeding after 24 h of clinical stability, increased with the number of risk factors present. However, the prediction ability for mortality and rebleeding of the Strate score were challenged in other studies [88,89]. Upper GIB scores such as GBS, AIMS-65, and pre-endoscopic Rockall (pRS) score have also been used for prediction of poor outcome in lower GIB. More recently, in 2017, Oakland specifically developed a new tool to detect safe discharge in low-risk patients with lower GIB [90]. It is based on different variables including age, sex, previous lower GIB history, digital rectal findings, heart rate, systolic blood pressure and hemoglobin. The external validation was performed in a large cohort of patients including different outcomes such as death, rebleeding, need of transfusion, therapeutic intervention, 28-day re-admission, and safe discharge. A cut-point of 8 or less was found to be adequate for safe discharge from the emergency department [90]. In addition, Oakland et al. performed a comparative study with other available scores to validate its own score. Mortality was better predicted with AIMS-65 (AUROC 0.78) and pRS (AUROC 0.75) [89]. Rebleeding was equally predicted with Oakland and GBS scores (AUROC 0.74) [89]. The need of blood transfusion was better predicted with the Oakland score (AUROC 0.92) [89]. Overall, good outcomes were achieved with the Oakland score. Today, GBS is a widely-used score and it has been shown to be useful for both upper and lower GIB [91]. More data from randomized studies are needed to establish the best score to predict outcomes in lower GIB.

### 3.5. Endoscopic Treatment

Endoscopic therapy in the colon and rectum can be extrapolated from that used in upper GIB including injection, mechanical, thermal, and topical therapy (Table 6). The information is limited, since no RCTs are available to compare different treatments in lower GIB [83].

High-risk stigmata in lower GIB are similar than those described in upper GIB such as active bleeding, both spurting and oozing, non-bleeding visible vessel, and adherent clots [84]. Endoscopic treatment should be guided for this hemorrhage stigmata.

Diverticular bleeding is usually self-limiting and as equal as other causes of lower GIB. Both mechanical and injection therapy is preferred to achieve hemostasis [83]. This approach allows low rates of early rebleeding, although late rebleeding is seen in up to 22% of patients [11]. Thermal therapy can be used with caution to avoid perforation risk, especially in the right colon. Band ligation has been described for treatment of active diverticular bleeding as another alternative, with a successful rate of 93% but with a high rebleeding rate up to 20% of cases [82,92,93]. Angiographic embolization or surgery are alternatives when endoscopic treatment cannot control the bleeding or instability hemodynamic is present.

Vascular angiectasia is another frequent cause of lower GIB. They are usually located in the right colon and non-direct thermal therapy with laser argon coagulation is the more typically used treatment [11]. Low-power setting should be programmed in the right colon to prevent perforation.

Hemospray^®^ could be used to stop diffuse sources of bleeding such as neoplasm, ischemic colitis, or inflammatory bowel disease. However, endoscopists must know that it has not been approved in some countries for its use in the lower gastrointestinal tract [83].

## 4. Small Bowel Bleeding

### 4.1. Risk Factors

Small bowel bleeding is becoming an entity on its own. Damage of the small bowel encompasses many pathophysiological ways, many of which remain unknown. Currently, there is a consensus that microbiota plays a fundamental role in the pathogenesis of many conditions including NSAIDS and ASA-associated enteropathy [4]. Different experimental studies in rats have shown that antibiotics prevent damage of the small bowel induced by NSAIDs [94,95]. One study in healthy volunteers showed that rifaximin, a microbiota modulator, was able to decrease the incidence of erosions and ulcers induced by diclofenac in the small bowel [96]. Moreover, probiotics may play a role in this point. A randomized clinical trial showed that Bifidobacteriumbreve Bif195 reduced the damage induced in the small bowel caused by Acetylsalicylic Acid when compared with placebo in healthy volunteers [97].

Drugs which modulate the microbiota, such as PPIs, have shown to influence the injury of the small bowel linked to NSAIDs and ASA. This hypothesis was suggested by Washio et al. in a randomized trial in Japan, who observed a high number of erosions in the small bowel of patients treated with a COX-2 inhibitor and PPI compared with patients receiving the COX-2 inhibitor plus placebo [98]. PPIs may cause dysbiosis by changing the gastric pH, although the mechanism is still unclear [99,100]. Drugs which perpetuate bleeding are another pillar related to small bowel bleeding. Antiplatelet and anticoagulant facilitate bleeding when any erosions or ulcer is present. Long-term aspirin use is related with erosive lesions along small bowel [101]. Aortic stenosis has been classically related with the presence of angiodysplasias (also known as Heyde syndrome). This relationship has not been observed with other valvular or ischemic heart diseases. Currently, the hypothesis is that the passage of blood through the defective valve causes the destruction of high molecular weight multimers of the von Willebrand factor, which causes their decrease and triggers a consequent tendency to bleeding in these patients [102].

### 4.2. Causes

Causes of small bowel bleeding include angiodysplasias, inflammatory bowel disease, Meckel’s diverticulum, erosions and ulcers related with drug use (mainly low-dose aspirin or NSAIDs), Dieulafoy’s lesion, and tumors (lymphoma, adenocarcinoma, carcinoid or polyp) [11].

Angiodysplasia is the most frequent cause of small bowel bleeding. About 90% of angiodysplasia bleeding cease spontaneously. However, their recurrence rate is high [103].

### 4.3. Diagnosis

If we exclude occult and often asymptomatic small bowel bleeding, melena is the most frequent form of clinical presentation of overt bleeding. However, red blood per rectum can also be observed if the bleeding flow is high.

Capsule endoscopy is the unique method to explore completely the small bowel mucosa, but no treatment can be applied with it. A randomized controlled trial showed that early capsule endoscopy was useful for detecting the source of bleeding in patients admitted for melena, hematochezia, or severe anemia in comparison with standard endoscopic investigation (64.3% vs. 31.1%; *p* < 0.01) [104].

Preparation before capsule endoscopy remains unclear. Some authors recommend only a low-fiber diet on the day before the procedure with clear liquids only in the evening and a 12 h fast [105]. However, some data suggest that ingestion of 2 L of polyethylene glycol solution prior to capsule endoscopy improves visibility of small bowel mucosa, and this is recommended for European Guidelines [105].

Fecal immunochemical test (FIT) is a good screening tool for colorectal cancer [106]. Some prospective studies have evaluated its usefulness to detect small bowel hemorrhage by capsule endoscopy in patients with normal upper and lower endoscopic studies. The data from these studies suggest that there is a correlation between a positive FIT and the detection of lesions in the small intestine, and suggests the usefulness of capsule endoscopy in patients with elevated FIT and normal colonoscopy. A review suggests that capsule endoscopy, if conducted early during or after the bleeding event, can identify the bleeding in at least one out of two patients [107]. The best candidates should be those with ongoing overt obscure bleeding or occult obscure bleeding. However, still there is not enough evidence through clinical trials and cost-effectiveness studies to widely recommend this test, and physicians should take into account other markers such as the presence of anemia and the amount of blood detected in the FIT to complete the investigation by capsule endoscopy [108,109,110]. ESGE guideline did not recommend the use of FIT to select patients for capsule endoscopy in the context of obscure GIB [111].

### 4.4. Endoscopic Treatment

Any endoscopic treatment of bleeding lesions located in the jejunum or ileum requires enteroscopy. This procedure is long and time consuming and requires endoscopists with experience. The single of double-balloon enteroscopy was developed to facilitate progression into the small bowel. Recently, another system based on a spiral has been created to help the endoscopist to explore small bowel lesions more quickly [112]. Thermal therapy assisted by laser-argon is commonly used in small bowel angiodysplasias. Endoscopic treatment is effective as initial therapy, but the rebleeding rate remains high (close to 34–45%) [113].

### 4.5. Medical Treatment

Withdrawing of aspirin, non-aspirin antiplatelet agents, or anticoagulants would be ideal in patients with small bowel GIB associated to these drugs. However, this is not always an option in patients with cardiovascular or thrombotic diseases. As previously commented, maintenance of aspirin treatment in CV patients decreases the risk of vascular events and death [114,115]. No specific studies have been focused on small bowel bleeding, and similar recommendations to those given for upper GI bleeding can be applied here. The diagnosis and treatment of the cause of the bleeding will facilitate the management of these patients. Otherwise, recurrence of the bleeding is the rule.

Medical treatment options are limited in these patients. A randomized trial was performed in Hong Kong and Japan to assess the efficacy of misoprostol (four times daily for eight weeks) compared to a placebo for treatment of low dose aspirin-induce injury in the small bowel. Capsule endoscopy was done before and after treatment, showing that misoprostol reduced the number of lesions caused by low-dose aspirin [116].

Octreotide is a somatostatin analog which has been used in small bowel bleeding due to angiodysplasia. Octreotide reduces the portal and mesenteric blood flow due to an inhibition of vasodilator peptides. Moreover, it increases the vascular resistance and improves platelet aggregation [113]. The number of patients evaluated with this drug in different studies is limited. Although some beneficial effects have been reported [116], more evidence from randomized controlled trials is needed. Data from OCEAN trial, which analyses the clinical effectiveness of octetride 40 mg compared to placebos taken for one year, will be available in the near future [103].

## 5. Conclusions

The management of acute nonvariceal GI bleeding is complex in an increasingly ageing population. The management is being facilitated by different technical advances in gastrointestinal endoscopy, which are essential to reduce rebleeding rates, need of embolization, and surgery. At the same time, available medical treatment provides a pillar in the management of these patients, whereas the appropriate management of antithrombotic drugs or NSAIDs which are usually taken by these patients is essential. The main objective is to reduce the morbidity and mortality associated with gastrointestinal bleeding from either the upper or the lower GI tract. However, this could increase as a result of aging populations and the consumption of antiplatelet and anticoagulant drugs.

## Figures and Tables

**Table 1 jcm-10-03045-t001:** Summary pre-endoscopy and post-endoscopy management.

Peptic Ulcer: Forrest	Initial Therapy	Endoscopy	Post-Endoscopy Management	Diet
Ia, Ib, IIa	80 mg IV PPI	Double endoscopic therapy	72 h of PPI– 8 mg/h in CP	Start oral ingestion after 24 h if no rebleeding
IIb	80 mg IV PPI	Consider therapy after clot removal	72 h of PPI– 8 mg/h in CP or PPI each 12 h	Start oral ingestion after 24 h if no rebleeding
IIc y III	40 mg IV PPI	No endoscopic treatment	Continue oral PPI	Start early ingestion

CP, continuous perfusion.

**Table 2 jcm-10-03045-t002:** Single antiplatelet agent management in non-variceal upper GIB.

Single Antiplatelet Agent	Presence of High-Risk Endoscopic Stigmata (FIa, FIb, FIIa)	Presence of Low-Risk Endoscopic Stigmata
Primary prophylaxis	Stop and re-evaluate the indication	Stop and re-evaluate the indication
Secondary prophylaxis	Stop and resume within 3 days or maintain the drug if hemostasia achieved	Continue drug use

**Table 3 jcm-10-03045-t003:** Dual antiplatelet therapy management in non-variceal upper GIB.

Dual Antiplatelet Therapy	Before Endoscopic Therapy	After Bleeding Control
Less than 6 months since CV event	Consult cardiologist (stop therapy if life-threatening bleeding). Try to maintain ASA.	Early resumption of therapy according to cardiologist
More than 6 months since CV event	Continue LDA and stop clopidogrel (or ticagrelor/prasugrel)	Resume: Clopidogrel: within 5 days Prasugrel: within 5 days Ticagrelor: within 3 days

**Table 4 jcm-10-03045-t004:** Vitamin-K antagonist anticoagulants management in non-variceal upper GIB.

Vitamin-K Antagonist	Before Endoscopic Therapy	After Hemorrhage Control
No high-risk patients	Stop the drug	Resume after 7 days
High-risk patient: Non-valvular AF with CHA2DS2-VASc > 3 Metallic mitral valve Prosthetic valve with AF <3 months after VTE Severe thrombophilia (protein C or S deficiency, antiphospholipid syndrome)	Stop the drug	Bridge therapy with LMWH and resume between 3–7 days

LMWH, Low-molecular-weight heparin.

**Table 5 jcm-10-03045-t005:** Direct oral anticoagulants management in non-variceal upper GIB.

Direct Oral Anticoagulants	Before Endoscopic Therapy	After Hemorrhage Control
No high-risk patients	Stop the drug	Resume within 5 days
High-risk patient: Non-valvular AF with CHA2DS2-VASc > 3 Metallic mitral valve Prosthetic valve with AF <3 months after VTE Severe thrombophilia (protein C or S deficiency, antiphospholipid syndrome)	Stop the drug	Resume within 48 h. Not necessary LMWH

**Table 6 jcm-10-03045-t006:** Summary of endoscopic therapies and their uses in upper and lower GIB.

Endoscopic Therapy	Classic	Indications	Modern Tools	Indications
Injection	Vasoconstrictors (epinephrine) and sclerosants	Peptic ulcer Dieulafoy’s Mallory-Weis Diverticular bleeding		
Mechanical	Endoclip	Peptic ulcer Diverticular bleeding Dieulafoy’s ulcer Mallory-Weiss tear	Over-the-scope (OTSC)	Peptic ulcer Diverticular bleeding Perforations
			Band ligation	Diverticular bleeding Esophageal varices GAVE Dieulafoy’s ulcer Hemorrhoid bleeding
			Endoscopic suturing	Refractory peptic ulcer Perforations
Thermal	Heater probe	Peptic ulcer	Radiofrequency ablation	GAVE
			Coagrasper	Bleeding secondary to ESD Peptic ulcer
	Non-contact laser argon	Angiectasias GAVE		
Topical			Hemospray^®^	Tumoral bleeding Peptic ulcer (with/after other therapy)
Others			Cryotherapy	Refractory GAVE

## Data Availability

Data sharing not applicable.

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
