# Peer review of "New Trends and Advances in Non-Variceal Gastrointestinal Bleeding—Series II"

_jcm, 2021, doi:10.3390/jcm10143045_

Round 1

Reviewer 1 Report

This is comprehensive review for diagnosis and treatment for overall GI bleeding. It is well-written, and it is helpful to will provide readers with the latest information on this issue. I have the following minor comments.

I would the authors to mention the contribution of small bowel bleeding to the overt-obscure gastrointestinal bleeding (OGIB), then to mention the usefulness of the simple clinical tests (e.g. fecal occult blood test) to create the opportunity for performing capsule endoscopy or double balloon to diagnose small bowel bleeding.

The authors needs to mention the ABC score, a new score to predict the mortality in upper and lower GI bleeding indistinguishably (Gut. 2021 Apr;70(4):707-716).

Author Response

Point 1: I would the authors to mention the contribution of small bowel bleeding to the overt-obscure gastrointestinal bleeding (OGIB), then to mention the usefulness of the simple clinical tests (e.g. fecal occult blood test) to create the opportunity for performing capsule endoscopy or double balloon to diagnose small bowel bleeding.

Response 1: Thank you for this interesting recommendation. We have included some studies which evaluate the presence of small bowel lesions, detected by capsule endoscopy, in patients with positive FIT and normal standard endoscopic studies.

Point 2: The authors need to mention the ABC score, a new score to predict the mortality in upper and lower GI bleeding indistinguishably (Gut. 2021 Apr;70(4):707-716).

Response 2: We have updated this paragraph and included the ABC score.

Reviewer 2 Report

An excellent review that comprehensively explains management for upper and lower gastrointestinal bleeding. From the basics to novel information that is not included in the guidelines because the evidence has not yet been established.

Major

Regarding the content of "Prognosis Score" on the second page, although the studies introduced by the authors have a high level of evidence, whether GBS is the optimal prognosis prediction score depends on the paper (cohort of the study).

GBS can be misunderstood as almighty, including the prediction of mortality, so it is better to review the description. Also, some aspects are not practical, such as many items and complexity.

Minor

The notation of Helicobacter pylori is good in italics (Helicobacter pylori).

The sentence on lines 123-124 of the third page is probably a mistake of "However, hemoglobin levels must be above 8 g/dl in patients with cardiac comorbidity (previous myocardial infarction, instability heart failure)". 

Author Response

Point 1: Regarding the content of "Prognosis Score" on the second page, although the studies introduced by the authors have a high level of evidence, whether GBS is the optimal prognosis prediction score depends on the paper (cohort of the study).

Response 1: We agree with the referee and have rewritten the score section to clarify this point.

Point 2: GBS can be misunderstood as almighty, including the prediction of mortality, so it is better to review the description. Also, some aspects are not practical, such as many items and complexity.

Response 2: We have rewritten the score section to include ABC score as the newest developed score to predict mortality in both upper and lower GIB. We have simplified the paragraph.

Point 3: The notation of Helicobacter pylori is good in italics (Helicobacter pylori)

Response 3: Thank you. We have modified the notation of the bacteria

Point 4: The sentence on lines 123-124 of the third page is probably a mistake of "However, hemoglobin levels must be above 8 g/dl in patients with cardiac comorbidity (previous myocardial infarction, instability heart failure)"

Response 4: Thank you for the suggestion. We have clarified the sentence in this way: “However, a higher cut-off level is recommended for transfusion in patients with cardiac comorbidities (e.g. previous myocardial infarction, heart failure, instability…), with transfusion being indicated when the hemoglobin level falls below 8 g/dl”